# Histone Deacetylases HD2A and HD2B Undergo Feedback Regulation by ABA and Modulate Drought Tolerance via Mediating ABA-Induced Transcriptional Repression

**DOI:** 10.3390/genes14061199

**Published:** 2023-05-30

**Authors:** Yongtao Han, Amira Haouel, Elisabeth Georgii, Santiago Priego-Cubero, Christoph J. Wurm, Daniel Hemmler, Philippe Schmitt-Kopplin, Claude Becker, Jörg Durner, Christian Lindermayr

**Affiliations:** 1Institute of Biochemical Plant Pathology, Helmholtz Munich, 85764 Oberschleißheim, Germany; hanyongtao515@163.com (Y.H.); ga62did@mytum.de (A.H.); elisabeth.georgii@helmholtz-muenchen.de (E.G.); c.wurm@posteo.de (C.J.W.); joerg.durner@helmholtz-munich.de (J.D.); 2Genetics, LMU Biocenter, Ludwig-Maximilians-Universität München, 80539 München, Germany; s.priego@biologie.uni-muenchen.de (S.P.-C.); claude.becker@biologie.uni-muenchen.de (C.B.); 3Research Unit Analytical Biogeochemistry, Helmholtz Munich, 85764 Oberschleißheim, Germany; hemmler@arcor.de (D.H.); philippe.schmittkopplin@helmholtz-munich.de (P.S.-K.); 4Chair of Biochemical Plant Pathology, Technische Universität München, 85354 Freising, Germany; 5Institute of Lung Health and Immunity, Comprehensive Pneumology Center, Helmholtz Munich, 85764 Oberschleißheim, Germany

**Keywords:** *Arabidopsis thaliana*, drought resistance, histone acetylation, plant-specific histone deacetylases, ABA

## Abstract

Histone deacetylation catalyzed by histone deacetylase plays a critical role in gene silencing and subsequently controls many important biological processes. It was reported that the expression of the plant-specific histone deacetylase subfamily HD2s is repressed by ABA in Arabidopsis. However, little is known about the molecular relationship between HD2A/HD2B and ABA during the vegetative phase. Here, we describe that the *hd2ahd2b* mutant shows hypersensitivity to exogenous ABA during the germination and post-germination period. Additionally, transcriptome analyses revealed that the transcription of ABA-responsive genes was reprogrammed and the global H4K5ac level is specifically up-regulated in *hd2ahd2b* plants. ChIP-Seq and ChIP-qPCR results further verified that both HD2A and HD2B could directly and specifically bind to certain ABA-responsive genes. As a consequence, Arabidopsis *hd2ahd2b* plants displayed enhanced drought resistance in comparison to WT, which is consistent with increased ROS content, reduced stomatal aperture, and up-regulated drought-resistance-related genes. Moreover, HD2A and HD2B repressed ABA biosynthesis via the deacetylation of H4K5ac at NCED9. Taken together, our results indicate that HD2A and HD2B partly function through ABA signaling and act as negative regulators during the drought resistance response via the regulation of ABA biosynthesis and response genes.

## 1. Introduction

Drought stress is one of the major environmental stresses, which leads to reduced photosynthesis and biomass accumulation, subsequently severely limiting the plants’ growth and agricultural productivity worldwide [1,2,3]. Plants have developed sophisticated strategies to cope with drought stress, such as improving water absorption by developing deeper and larger root systems, reducing water loss by regulating stomatal closure, and increasing the level of antioxidants, moreover, phytohormones and transcription factors also help plants survive from drought by regulating the expression of stress-induced genes [3,4,5]. 

When plants suffer from drought stress, the gene expression profile and metabolic pathways were reprogrammed and distinct compounds, such as late embryogenesis abundant (LEA) proteins and the phytohormone abscisic acid (ABA) were produced to help plants better cope with the deleterious environmental conditions [4,6]. The phytohormone ABA is a key plant hormone involved in plant adaptation to drought stress. Numerous genes involved in drought stress responses and tolerance are regulated through the ABA signal transduction pathway [7,8,9]. Under drought stress, roots sense the dehydration signals and transfer the signals to leaf mesophyll cells to drive ABA production and trigger the plant drought response [10]. The 9-cis-epoxycarotenoid dioxygenase (NCED) catalyzes 9-cis-epoxycarotenoid cleavage reaction and is believed as the rate-limiting step in ABA biosynthesis. *NCED3* and *NCED9* are believed to be key genes in ABA synthesis under drought-stressed conditions in Arabidopsis since their transcription is induced by drought stress [11]. NCED overexpressing Arabidopsis plants accumulated more ABA and displayed enhanced drought resistance [11,12]. Since ABA acts as a signaling molecule, activation of ABA signal transduction is also crucial for the enhancement of dehydration tolerance. Overexpression of ABA receptor and signal transduction components or repression of negative regulators of ABA signaling enhanced the plants’ dehydration tolerance [13,14,15,16,17,18,19,20]. ABA is recognized by the ABA receptor and the ABA signal is transducted via transcription factors to promote the expression of ABA-responsive genes [8], subsequently resulting in physiological reactions, such as stomatal closure, Reactive Oxygen Species (ROS) accumulation, and altered root architecture [21,22,23,24]. 

Reversible histone acetylation, which is controlled by the opposing action of histone acetyltransferases (HATs) and histone deacetylases (HDACs), is one of the most studied post-translational modifications (PTM) in epigenetics. Histone acetylation dynamics are directly connected with transcriptional activation and silencing in eukaryotes [25] and gene silencing via histone deacetylation is a universally conserved epigenetic regulation system in eukaryotes [26,27]. HDACs are ubiquitous in eukaryotes and an increasing number of HDAC genes have been identified and characterized in a variety of plant species such as maize, Arabidopsis [25], barley [28], potato [29], grape [30], and tobacco [31]. In Arabidopsis, 18 HDACs genes are grouped into three subgroups: RPD3-family (reduced potassium deficiency 3), HDT-family (plant-specific histone deacetylases, HD2s), and sirtuin-family, [25]. An increasing amount of evidence revealed that HDACs are involved in the regulation of plant stress response as well as plant development. High salinity, cold stress, and exogenous ABA treatment caused a rapid and transient up-regulation of H3 and H4 acetylation in Arabidopsis and tobacco [32]. At the coding regions of drought stress-responsive genes, PTMs of the N-terminal tail of histone H3 are altered under drought stress [33], e.g., the acetylation level of H3K9, H3K18, H3K27, and H4K5 were enhanced and kept elevated for consecutive droughts in rice [34]. In Arabidopsis, RPD3 subfamily members HDA9 and HDA15 are required for plant thermal response and function as a suppressor of negative regulators of ABA signaling [35,36,37,38]. HDA6 and HDA19 another 2 RPD3 members, function redundantly in modulating seed germination and salt stress response [39]. Studies demonstrated that Arabidopsis HDA6 and HDA19 redundantly contribute to the post-germination development directly or indirectly via repression of embryo-specific gene function [40]. HDA19 and HDA6 and HDA15 also have been shown to function in seed germination, flowering, and leaf development [41,42,43,44,45]. Expression of all four members of the plant-specific histone deacetylase was repressed by exogenous application of ABA and salt [46] suggesting a negatively regulatory role of HD2s in stress response. Surprisingly, overexpression of HD2A, HD2C, and HD2D enhanced tolerance to drought and salt stress in Arabidopsis [47,48]. Moreover, silencing of HD2C conferred an ABA-sensitive phenotype and decreased tolerance to salt stress and heat stress of the mutant line [46,49,50]. These observations suggest an activating and inactivating function of HD2s in ABA signalling and a highly coordinated interaction between HD2s function, ABA signalling and stress response. 

Besides function in stress response, HD2C and HD2D are involved in regulating Arabidopsis root growth and flowering time [51]. Furthermore, HD2A and HD2B function is required for root and rosette development, plant reproductivity, and seed germination [52,53,54]. In this study, we characterized the molecular function of the plant-specific histone deacetylases HD2A and HD2B of Arabidopsis in drought stress response mediated through ABA-induced transcriptional repression. The expression of HD2A and HD2B was repressed by ABA treatment, and genetic and molecular analysis indicates that HD2A and HD2B function redundantly in regulating drought stress via repressing ABA-induced stress-responsive genes. Loss of and reduced function of HD2A and HD2B, respectively, leads to an ABA-hypersensitive phenotype. In addition, NCED9 expression, required for ABA synthesis, is upregulated in *hd2ahd2b* mutant resulting in enhanced accumulation of ABA and ROS in comparison to WT. Moreover, after polyethylene glycol (PEG) treatment, ROS accumulation was further increased in *hd2ahd2b* leaves compared to WT, which might contribute to the observed reduced stomata aperture in the *hd2ahd2b* mutant. Transcriptome analysis revealed that transcription ABA-response genes are significantly upregulated in *hd2ahd2b* plants. Interestingly, the differentially expressed genes of the *hd2ahd2b* mutants are significantly overlapping with the differentially expressed genes of ABA-treated WT plants, and these overlapped genes are shown quite similar genes expression patterns between *hd2ahd2b* plants and WT plants treated with ABA, suggesting that HD2A and HD2B play an essential role in ABA signal transduction pathway. Moreover, more than half of the overlapped gene expression does not change in *hd2ahd2b* plants after being treated with ABA, leading to the conclusion that the expression of these ABA-response genes is HD2A and HD2B dependent. We demonstrated that on one side, HD2A and HD2B are responsible for deacetylating H4K5ac in certain ABA-responsive genes, and on the other side both plant-specific histone deacetylases are involved in deacetylating genes involved in ABA synthesis. 

Taken together, these data suggest that HD2A and HD2B are involved in the regulation of the drought stress response via the repression of ABA-activated genes by binding to their coding region and deacetylating the H4K5ac histone mark.

## 2. Results

### 2.1. Loss of HD2A and Reduced Function of HD2B Results in Hypersensitivity towards ABA

A previous study showed that the expression of *HD2A* and *HD2B* was repressed by the exogenous application of ABA [46]. We also observed a reduced expression of *HD2A* and *HD2B* after the treatment of 10-day-old Arabidopsis wild-type (WT) seedlings with 100 µM ABA (Figure 1A). To untangle the plausible role of HD2A and HD2B in the ABA response, we analysed the T-DNA insertion mutant *hd2a* and *hd2b* (Col-0 background). *hd2a* mutant is a null mutant, whereas *hd2b* is a knockdown line with an 80% reduction of *HD2B* transcripts [53]. Previous research showed that HD2A and HD2B are related in terms of expression and function redundantly during the development of roots and the reproductive system, and the null mutant of *hd2ahd2b* is lethal [52,53]. Therefore, we generated the *hd2ahd2b* double mutant line by crossing the *hd2a* null mutant and *hd2b* knockdown mutants. The germination ratio, root growth, and cotyledon development of the *hd2a* and *hd2b* single mutants as well as of the double mutant line were analyzed. After-ripened seeds of *hd2a*, *hd2b*, *hd2ahd2b,* and WT plants were germinated on 1/2 MS medium with or without 0.5 µM ABA under short-day conditions. As shown in Figure 1B–E, the germination ratio, root growth, and cotyledon development of both *hd2a* and *hd2b* single mutants are not significantly different from WT. In contrast, the *hd2ahd2b* double mutant displayed hypersensitivity towards ABA in comparison to WT (Figure 1B-E). While nearly all WT seeds germinated after 3 days, *hd2ahd2b* seeds need more than 9 days for complete germination (Figure 1B). In the presence of 0.5 µM ABA, the delay of germination is even more pronounced (Figure 1C). While almost all of the WT seeds germinated after 3 days of incubation, the seeds of the *hd2ahd2b* double mutant showed no germination at that time point, and only about 40% of the seeds germinated after 7 days of incubation (Figure 1C). Additionally, the primary root lengths and percentage of green cotyledons of WT and *hd2ahd2b* line grown in the presence of different ABA concentrations (0, 0.1, and, 0.2 µM) were analyzed 7 days after germination. The root length of the *hd2ahd2b* seedlings is significantly shorter than the roots of WT seedlings under control conditions and root growth is severely inhibited in the presence of ABA (Figure 1D). Moreover, exogenous ABA also significantly reduced the percentage of green cotyledons of *hd2hd2b* seedlings even at very low concentrations of ABA (Figure 1E). These findings suggest that the loss function of HD2A and reduced function of HD2B enhance sensitivity to exogenous ABA in Arabidopsis, suggesting an interconnectedness between HD2A and HD2B function and ABA signaling. 

### 2.2. HD2A and HD2B Function Is Essential for the Repression of ABA-Responsive Genes

Since ABA is down-regulating the expression of HD2A and HD2B, we assumed that both plant-specific histone deacetylases negatively regulate the expression of ABA-responsive genes. Therefore, we performed an RNA-sequencing (RNA-seq) analysis of ten days old WT and *hd2ahd2b* seedlings treated with or without 100 μM ABA for 6 h. Differentially expressed genes were defined based on a threshold of at least 2-fold change (*p*-value < 0.05). Compared to the WT control sample, 3920 differential expressed genes (DEGs) (including 2258 up-regulated genes and 1662 down-regulated genes) were identified in WT seedlings treated with ABA, whereas 2492 DEGs (including 1720 up-regulated genes and 772 down-regulated genes) were identified in *hd2ahd2b* seedlings without ABA treatment (Appendix A [55]). Since ABA down-regulating the expression of HD2A and HD2B, we expected a similar expression pattern between “WT/ABA vs. WT/control” and “*hd2ahd2b*/control vs. WT/control”. Indeed, we observed an overlap of 706 up-regulated and 314 down-regulated genes (Figure 2A, Appendix A). Gene Ontology enrichment analyses of the overlapping genes revealed that the up-regulated genes are enriched in genes related to response to water deprivation, response to osmotic stress, response to ABA, and response to water (Appendix A), whereas the down-regulated genes are enriched in genes related to S-glycoside biosynthetic process, sulfur compound biosynthetic process, Glucosinolate biosynthesis from methionine, and secondary metabolite biosynthetic process (Appendix A). 

Next, we compared the expression of these overlapped genes with similar expression patterns (706 up-regulated genes, and 314 down-regulated genes) in *hd2ahd2b* mutants with and without ABA treatment (Figure 2B, Appendix A), and we found that about 56.9% (399 of 706 up-regulated genes) of the activated overlapped genes and 64.5% (202 of 314 down-regulated genes) of the repressed overlapped genes by ABA in WT plants did not respond to ABA treatment in the *hd2ahd2b* mutants (Figure 2B, Appendix A), concluding that the ABA-induced transcription changes of these genes are dependent on HD2A and HD2B. In addition, the remaining 41% of overlapping genes (419 (307 up-regulated and 112 down-regulated) of 1020 genes) show similar gene expression patterns both in “WT/ABA vs. WT/control” and “*hd2ahd2b*/ABA vs. *hd2ahd2b*/control” (Figure 2C, Appendix A), represent those ABA-responsible gene expressions independent of HD2A and HD2B. Gene ontology analysis showed that the GO term “response to ABA”, “response to stress”, “response to oxygen-containing compound”, and “response to abiotic stimulus” were enriched among the ABA-activated genes that were dependent on HD2A/B (Figure 2D, Appendix A). These results suggest that HD2A and HD2B are involved in the regulation of genes repressed by ABA, and these genes potentially play a crucial role in the stress response.

### 2.3. Under Control Conditions, HD2A and HD2B Repress ABA-Responsive Genes via Histone Deacetylation

Although HD2A and HD2B have been recognized as HDACs, their exact biochemical function in Arabidopsis is still unknown [56,57]. We found that the loss of HD2A and reduced function of HD2B caused a 45% reduction in total HDAC activity compared to the WT [55], highlighting the crucial role of both proteins in HDAC activity. To further examine the histone deacetylating functions of HD2A and HD2B, global acetylation levels of H4, H4K5, and H3K9 in 4 weeks old rosette leaves of WT and *hd2ahd2b* mutant were quantified by immunoblotting. 

The H4K5ac level was significantly enhanced in the *hd2ahd2b* leaves compared to the WT, but no significant difference was observed in the H3K9ac and H4ac levels (Figure 3A), suggesting that HD2A and HD2B regulate H4K5ac levels in leaves of 4 weeks old plants. According to the results of the RNA-seq analysis, the repression of 399 ABA-responsive genes under control conditions depends on HD2A and HD2B function (Figure 2C). Histone deacetylases can repress gene transcription by binding to the corresponding gene loci and deacetylating histone lysine/arginine residues. RNA-seq and ChIP-seq are complementary approaches usually used for the comprehensive analysis of plant transcriptional regulatory mechanisms [58,59]. To correlate transcriptional changes in the *hd2ahd2b* mutant with HD2B binding, we re-analyzed the publicly available HD2B binding ChIP-seq data [60]. We compared the group of genes up-regulated in *hd2ahd2b* (in comparison to WT) with the group of genes, to which HD2B was bound to. We found that 466 of the 1720 genes up-regulated in *hd2ahd2b* (27.1%) are also binding HD2B (Appendix A). A Gene Ontology enrichment analysis revealed that this group of genes is enriched in genes related to response to abiotic stimulus (*p* = 1.2 × 10^−6^), response to stress (*p* = 3.2 × 10^−5^), and response to ABA (*p* = 1 × 10^−4^) (Appendix A). In detail, 30 genes were related to ABA response (Appendix A), and 100 genes were involved in stress response (Appendix A). Subsequently, we further analyzed whether HD2B directly binds to certain ABA-responsive gene loci. Some of the representative ABA response genes from Appendix A (ABI4, AHG1, RAB18, Em1, and ABI3), and two other genes from the *hd2ahd2b* transcriptome upregulated gene list (LTP4 and XERO1) were selected, and analyzed by RT-qPCR and ChIP-qPCR. RT-qPCR confirmed the enhanced transcription of these genes in *hd2ahd2b* plants as well as after ABA treatment (Figure 3B). Moreover, we performed ChIP-qPCR with 10-day-old seedlings of WT and *hd2ahd2b* complemented with *35Spro: HD2B-GFP* using an anti-GFP antibody. An intergenic region between AT5G43175 and AT5G43180 was selected as the negative control. Fragments corresponding to the coding region of the selected genes were significantly enriched confirming HD2B-GFP binding (Figure 3C). No significant enrichment was detected in the negative control. These results demonstrate that HD2B affects histone acetylation and transcription of these ABA-responsive genes. It was reported that both HD2A and HD2B could affect plant reproductive development by regulating the transcription of rDNAs and rRNA processing-related genes through histone acetylation [52], so we wanted to know whether HD2A also acts similarly to HD2B on these ABA-responsive genes. Therefore, we re-analyzed published HD2A-GFP ChIP-seq data [52] and revealed that HD2A-GFP is also enriched in these genes (Figure 4). In conclusion, both HD2A as well as HD2B act on these ABA-responsive genes. Taken together, these results demonstrate that HD2A and HD2B function as repressors of numerous ABA-responsive genes, possibly via the removal of histone acetylation at these gene loci.

### 2.4. Loss of HD2A and Reduced Function of HD2B Affects ABA Content, ROS Accumulation, and Stomatal Closure

Besides the up-regulated expression level of HD2A/B-dependent ABA-responsible genes in the *hd2ahd2b* plants, the enhanced expression level of HD2A/HD2B-independent ABA-induced genes (Appendix A) was analyzed, especially the enhanced ABA synthesis gene NCED9 expression level.

Figure 5B may indicate a higher ABA content of *hd2ahd2b* plants under control conditions. So, we determined the endogenous ABA content of 4 weeks old unstressed WT and *hd2ahd2b* plants. Indeed, the *hd2ahd2b* plants had a significantly higher ABA content than WT plants (Figure 5A). Since enhanced histone acetylation is related to gene expression, the increased accumulation of ABA in the *hd2ahd2b* plants might be rather a result of enhanced ABA synthesis than diminished ABA degradation. Therefore, we analyzed the expression level of the main genes involved in ABA synthesis in rosette leaves of four weeks old unstressed WT and *hd2ahd2b* plants (Figure 5B). Among the analyzed genes, NCED9 is the only gene whose expression is significantly upregulated in *hd2ahd2b* compared to WT (Figure 5C). Additionally, ChIP-qPCR and ChIP-Seq data show that both HD2A and HD2B are enriched in the genomic region of NCED9, and loss of HD2A and reduced function of HD2B caused hyperacetylation of H3K9 and H4K5 at this locus (Figure 5D–F). These results demonstrate that the up-regulation of NCED9 in *hd2ahd2b* plants is correlated with enhanced histone acetylation.

ABA is considered to be the major phytohormone that combats drought stress. The exogenous application of ABA increased plants’ drought tolerance [7,61,62]. One of the biological activities of ABA during drought stress is to induce the accumulation of ROS and expression of stress-responsive genes. Moreover, ROS accumulation subsequently triggers stomatal closure to reduce transpiration and finally improve the drought tolerance of plants [22,63,64].

We found that the *hd2ahd2b* plants accumulated more ROS than WT plants even under unstressed conditions and the difference was more obvious after PEG irrigation (Figure 6A). Consistent with the ROS accumulation, the *hd2ahd2b* plants displayed an enhanced stomatal closure in comparison to WT plants and the difference increased after the supply of ABA and PEG (Figure 6B), while no difference in stomata density was observed between them (Figure 6C). All these results suggest that the *hd2ahd2b* plants might be more drought tolerant than WT plants.

### 2.5. Loss of HD2A and HD2B Confers Drought Resistance in Arabidopsis

Here, both *hd2a* and *hd2b* single mutants show the same ABA/salt sensitive and drought-resistant phenotype, when compared to WT (Figure 1C, D, E, Appendix A). To investigate the drought tolerance of *hd2ahd2b* plants, a limited water supply was simulated via irrigating four weeks old WT and *hd2ahd2b* plants with 25% PEG 6000. After 24 h, WT plants were completely wilt, while the *hd2ahd2b* plants did not show any visible drought stress phenotype (Figure 7A). Additionally, to quantify the plant drought stress response, we performed a drought stress experiment with *hd2ahd2b* and WT plants using the automated phenotyping platform PlantScreen TM Compact System (PSI, Czech Republic), which enables non-invasive, objective, impartial, and convenient measurement of morphological and physiological parameters. 60 WT and 60 *hd2ahd2b* plants were grown under long-day conditions and at an age of 2 weeks, watering was stopped for half of the plants, while the other half of the plants were well-watered during the whole growth period. Loss of water was monitored every two days via weighting the pots (Figure 7B). Both WT and *hd2ahd2b* plants lost 60 mL of water after 30 days without watering demonstrating that there is no significant difference in soil water content between WT and *hd2ahd2b* plants (Figure 7B). The control plants were watered every two days up to a total weight of 180 g. The Red Green Blue (RGB) imaging and kinetic chlorophyll fluorescence imaging of control and drought-stressed plants were performed every two days with the PlantScreen TM Compact System to monitor plant growth and photosynthetic performance. Finally, the morphological and physiological parameters including the Chl fluorescence parameter (F_v_/F_m_), rosette areas, and perimeter were analysed. The maximal PSII quantum efficiency (F_v_/F_m_) is an appropriate index to assess the drought stress-mediated photoinhibition of plants which could reflect the fitness of plants, and the plants are considered to be healthy and non-stressed if the F_v_/F_m_ value is around 0.8 [65,66]. A decrease in F_v_/F_m_ ratio reflects the damage of the PSII center when plants suffered from stress, and a higher F_v_/F_m_ ratio under stress conditions is considered to be more stress resistant [67,68]. After 23 days without watering, the leaves of *hd2ahd2b* remained green and dense, whereas those of the WT plants were etiolated and wilt (Figure 7C). In consistence with the observed lower drought tolerance, the F_v_/F_m_ ratio of the WT plants decreased 10 days earlier (22 days without watering) than the F_v_/F_m_ ratio of the *hd2ahd2b* plants (32 days without watering), indicating the *hd2ahd2b* mutant plants withstood the drought stress conditions significantly better than the WT plants (Figure 7C–E). These results suggest that under the same drought conditions, the *hd2ahd2b* mutant has a higher water use efficiency and drought tolerance ability than WT plants as WT plants’ maximal PSII quantum efficiency is more vulnerable to drought stress than that of *hd2ahd2b* plants. Additionally, we analyzed the leaf area of WT and *hd2ahd2b* plants in response to progressive drought stress, since insufficient water supply inhibits growth and finally results in the wilting/shrinking of the leaves. The results showed a growth reduction under drought conditions in both *hd2ahd2b* and WT plants (Figure 7F). In general, WT plants are growing faster than *hd2ahd2b* plants. This might be at least partly related to the delayed germination of *hd2ahd2b* seeds. However, on day 28 of the experiment, the well-watered control plants of WT (2800 mm^2^) and *hd2ahd2b* (2600 mm^2^) have a similar leaf area. Surprisingly, after seven days without watering the *hd2ahd2b* plants showed already a significantly smaller leaf area than the corresponding well-watered control plants, whereas a significant decrease in the leaf area of WT plants was observed after 14 days without watering (Figure 7F). The early growth reduction of the *hd2ahd2b* plants might be related to their drought stress tolerance.

Besides the morphological and physiological analyses, we analysed the expression level of marker genes for stress response (DREB2A, RD22, and RD29A). The transcripts of these genes were quantified by qRT-PCR in WT and *hd2ahd2b* plants grown under drought stress and control conditions. The expression level of all analysed marker genes was up-regulated under drought stress conditions and significantly higher in *hd2ahd2b* plants in comparison to WT plants (Figure 7G). Consistent with the RNA-seq data, the expression level of RD22 under control conditions is already significantly higher in *hd2ahd2b* plants in comparison to WT plants (Appendix A, Figure 7G). All these results indicate that loss of HD2A and HD2B function enhanced the Arabidopsis drought resistance by promoting the expression of ABA-responsive and stress-related genes. 

### 2.6. Loss of HD2A and HD2B Caused a Pleiotropic Phenotype

In our results, we also found that, besides a drought tolerance phenotype, the *hd2ahd2b* plants displayed a pleiotropic phenotype. Compared to *hd2a* and *hd2b* single mutant and WT plants, *hd2ahd2b* plants show significantly smaller rosette diameters (Figure 8A,C). Moreover, in comparison to WT plants, *hd2ahd2b* plants have a greenish-yellowish leaf color due to reduced chlorophyll b content (Figure 8B), pointed leaves (Figure 8D), a late flowering phenotype (Figure 8E), more lateral shoots (Figure 8F), and the seed development of *hd2ahd2b* line is also disturbed (Figure 8G). And all these phenotypes are reminiscent of the phenotype of a ribosome-biogenesis deficient phenotype [69,70]. Additionally, HD2A, HD2B, and HD2C could form homo-oligomer and be involved in plant development by affecting ribosomal RNA (rRNA) and snoRNA processing and methylation [52,55,71], A former colleague found that the NOP56 and FIB2 (the core members of box C/D snoRNA-associated proteins complex which multifunction in snoRNA processing, 2′-O-methylation of rRNA, and snoRNA transport to the nucleolus [72]) are putative interaction partner proteins of HD2C by Co-IP, which was confirmed by BIFC (Appendix A). That indicates that HD2A and HD2B may also work as part of this protein complex. Moreover, our BiFC results confirm that both HD2A and HD2B could interact with NOP56 and FIB2 (Figure 9), which may extend our understanding of the molecule functions of HD2A and HD2B in drought resistance through rRNA processing and methylation. Taken together, these findings indicate that HD2A and HD2B may play a critical role in coordinating plant drought tolerance and development.

## 3. Discussion

To date, increasing studies have shown that HD2s can mediate transcriptional repression by modifying histone [52,53], and certain HD2s are involved in general developmental processes [52,73,74] and plant responses to abiotic stresses [46,47,50,75]. This suggests that plant-specific histone deacetylases play a globally important role in plant development and response to environmental stresses. In this study, we present that HD2A and HD2B negatively affect Arabidopsis drought resistance by acting as negative regulators mainly via modulating the ABA signaling by repressing ABA biosynthesis and ABA-responsive genes, and partly through ABA independent pathways, e.g., specific plant morphology (small leaves).

### 3.1. HD2A and HD2B Serve as Negative Regulators of Plant Drought Resistance

Since the *hd2ahd2b* null double ko-mutant is lethal [52], we generated the *hd2ahd2b* mutant by crossing the HD2A knock-out and HD2B knock-down line. The HD2B T-DNA insertion line displays a reduction of approx. 80% of HD2B transcripts and showed a similar phenotype as the CRISPR/cas knockout line [52,71], concluding that the *hd2b* knock-down mutant is suitable for functional gene analysis.

Accumulating evidence has shown that epigenetic modifications mediating transcriptional regulation play pivotal roles in plants’ responses to environmental challenges [76,77,78]. Previous studies have reported that the expression of plant HDACs was differentially regulated when suffering from drought stress, subsequently, influencing the histone acetylation levels of drought-related genes and changing the drought tolerance characteristics of plants [36,39,75]. In Arabidopsis, loss of HDA19 could enhance plant drought resistance [79], whereas the drought tolerance was reduced in *hda9* mutants [36]. The Arabidopsis HD2A, HD2C, and HD2D were described as positive regulators of drought resistance since overexpression of these three genes played enhanced tolerance to drought stress when compared with WT plants [47,48,50]. While in another study, the silencing of HD2D represses enap1enap2myb44 drought-sensitive phenotype, indicating HD2D negatively regulates plant drought response [80]. The inconsistent outcomes for HD2D may be due to the different backgrounds of the Arabidopsis lines. Commonly, the ABA-induced genes play positive roles during drought resistance [8,81], and the transcription levels of all four Arabidopsis HD2s were repressed after being treated with 100 μM ABA (Figure 1A) [46], which suggested a potentially negative role in drought resistance, consistence with that, we found HD2A and HD2B function redundantly act as negative regulators of drought resistance in Arabidopsis, as *hd2ahd2b* plants displayed enhanced drought resistance compared to WT and single knockout lines under both natural and PEG-induced drought conditions (Figure 1, Figure 7, and Appendix A). Usually, increasing root water uptake from the soil and reducing water loss by closing stomata are the two main ways for plants to cope with drought stress [4,5]. However, *hd2ahd2b* shows shorter and fewer roots than WT plants (Figure 1D) [53], which indicates less water uptake, implying that HD2A and HD2B may regulate drought stress by regulating stomatal dynamics, and that coping with the stomatal aperture in the result (Figure 6B).

### 3.2. HD2A and HD2B Regulate Drought Stress Responses through an ABA-Dependent Pathway

ABA plays a key role in plant stress responses. Drought stress induces the accumulation of ABA, which is an endogenous messenger that transmits stress signals and regulates the expression of various stress-responsive genes and initiates many adaptive responses by a complex regulatory network [8,9,82]. In our study, *hd2ahd2b* plants show hypersensitivity to ABA, and the endogenous ABA content and expression levels of several ABA-responsive genes were changed in *hd2ahd2b* mutant compared with WT plants (Figure 1, Figure 3B, Figure 5A and Figure 7G), based on these results, we proposed that HD2A and HD2B are involved in ABA-mediated plant drought resistance. 

It is reported that over-accumulation of ABA induces ROS production, which in turn, results in the declined stomatal aperture [24,83,84]. The endogenous ABA content and ROS accumulation were both higher in *hd2ahd2b* plants in comparison to WT plants under control conditions (Figure 5A and Figure 6A). The enhanced accumulation of ABA is most likely due to the enhanced NCED9 expression, as only NCED9 expression was upregulated in all the ABA synthesis genes we tested (Figure 5B). Interestingly, the H4K5 and H3K9 acetylation levels of NCED9 were elevated in the *hd2ahd2b* mutant, and ChIP-Seq as well as ChIP-qPCR results indicate that both HD2A and HD2B could directly bind to the NCED9 gene (Figure 5C–E). These results suggest HD2A and HD2B affect the accessibility of NCED9 for the transcription machinery by regulating its acetylation level, which decreases or increases the expression level and subsequently promotes ABA synthesis. 

We demonstrate that the expression level of HD2A and HD2B was reduced by the exogenous application of ABA (Figure 1A) [46], suggesting that a negative feedback loop between ABA accumulation and HD2A/HD2B repression coordinates the response to drought stress in Arabidopsis. The accumulation of ROS in the *hd2ahd2b* mutant might be partly a result of the enhanced ABA level and partly independent of ABA, since we observed enhanced expression of certain (e.g., PER44, PER34) in the transcriptome of *hd2ahd2b* mutant (Appendix A). ROS are important second messengers involved in regulating physiological and developmental processes in plants [85,86,87]. The enhanced ROS level in *hd2ahd2b* plants might be responsible for the inhibited stomata opening in these plants, which is further pronounced after ABA or PEG treatment (Figure 6A,B). Since the stomatal density is not significantly different between *hd2ahd2b* and WT plants (Figure 6C), these results indicate that *hd2ahd2b* plants might lose less water via transpiration than WT plants.

### 3.3. HD2A and HD2B Repress the Expression of Certain ABA-Responsive Genes

Epigenetic mechanisms such as histone modifications are involved in the regulation of gene transcription mainly by modulating the chromatin structure, and some of those genes are ABA and drought-responsive, which play crucial roles in plants’ environmental stress responses [34,88,89,90]. Under dehydration condition, the enrichment of HDT4 is reduced on the promoter region of UGT74E2, RD28, and DREB2C, which enable the deposit of H3K27ac and the elevation of those drought-responsive genes, thus, plants are drought resistant [80]. H3K9ac level of RD20, RD29A, and RD29B has been found to enrich rapidly in these gene regions in response to drought stress [33]. HD2s have long been recognized as plant-specific histone deacetylases, which generally function as transcription repressors [52,53]. Here, we found that HD2A and HD2B mediate ABA-induced transcriptional repression in Arabidopsis. The hypersensitive phenotype of *hd2ahd2b* mutants to ABA indicates that HD2A and HD2B negatively regulate the ABA responses (Figure 1C–E and Figure 3B). The very similar gene expression pattern between *hd2ahd2b* double mutant in comparison to ABA-treated WT plants strongly indicated a link between HD2A/HD2B function and the ABA signaling pathway (Figure 2B). Moreover, RNA-seq data also shows that certain ABA-induced genes are HD2A/HD2B-dependent (Figure 2C). The GO analysis shows that, those HD2A/HD2B dependent ABA-induced genes enriched in GO terms associated with drought processes, such as “response to stress”, “response to ABA”, and “response to water” (Figure 2D). Exogenous ABA treatment can enrich histone acetylation of some ABA response genes in Arabidopsis and cause rapid up-regulation of H4 acetylation in tobacco cells [32,91]. In the *hd2ahd2b* mutant H4K5ac level was specifically enhanced (Figure 3A) suggesting a link between the HD2A/HD2B mediated dynamic changes of H4K5ac level and regulation of ABA-induced gene expression. Furthermore, the ChIp-qPCR and ChIP-Seq assay show that both the HD2A and HD2B could directly bind to the coding regions of distinct HD2A/HD2B-dependent ABA-inducible genes and might catalyze H4K5 deacetylation on these loci (Figure 3C and Figure 4). These results highlight that HD2A and HD2B are essential for the repression of ABA-inducible genes via histone deacetylation. 

### 3.4. HD2A and HD2B Function in Drought Tolerance by Affecting ABA-Independent Signaling Pathways and Ribosome Biogenesis

Genes functioning in drought response can be broadly classified into ABA-dependent and ABA-independent genes [92]. Besides regulating drought responses through an ABA-dependent pathway, HD2A/HD2B might also be involved in drought responses via ABA-independent mechanisms. For E.g. transcription of the ABA-independent drought-responsive genes DREB2A and RD29A were enhanced in *hd2ahd2b* under drought stress conditions (Figure 7G), and, transcriptome analysis shows the GO term “response to stress” is also found in HD2A/HD2B dependent but ABA-independent genes (Appendix A).

Besides the level of transcription, plants respond to environmental changes also on the levels of translation [93]. The ribosome is a complex molecular machine made of ribosomal RNA and ribosomal proteins and is responsible for translation. Growing evidence suggests that the integrity of the ribosome is crucial for organisms’ normal development and abiotic stress response [69,70]. In bacteria, ribosome hibernation caused slowed down protein synthesis rate and enhanced resistance to adverse chemicals or conditions of the cell [94]. Ribosomal protein gene deletion in *saccharomyces cerevisiae* strains shows slower growth and enhanced resistance to endoplasmic reticulum stress [95]. In Arabidopsis, APUM23, a nucleoli protein involved in ribosome biosynthesis, is essential for salt sensitivity through and downstream of the ABA signaling pathway [96], and loss of rRNA processing protein 7 (Rrp7) shows a pleiotropic rosette phenotype and partial infertility, and ABA hypersensitivity in seedlings [97]. Interestingly, *hd2ahd2b* plants display a typical ribosomal mutant phenotype (Figure 8), which was already in part reported previously and quite similar to the phenotype of *apum23* (slow growth and pointed leaves) [52,69,70,98]. The transcriptome analysis revealed that the expression of ribosome biogenesis proteins both in *hd2ahd2b* and *apum23* mutant is affected [55,98]. Additionally, the previous study attributes the pleiotropic phenotype of the *hd2ahd2b* line to a result of the deregulation of ribosome biogenesis as well as other nucleolar functions [52], while the interaction of HD2A/HD2B with FIB2 and NOP56 (Figure 9), two core proteins of Arabidopsis C/D box snoRNP complex involved in RNA methylation, suggest that HD2A/HD2B may play a critical role in coordinating stress responses and plant development by dynamically regulating HDACs-mediated rRNA post-transcriptional modification, which broadens and deepen the understanding of HD2A and HD2B in ribosome biogenesis, but the molecular mechanisms still need to be further studied. Taken together, these results indicate that HD2A/HD2B might regulate the transcription of ribosome biogenesis-related genes and in this way affect the ribosome composition and their post-transcriptional modulation to guarantee normal development of Arabidopsis and involvement in stress response. 

Water is crucial for plant growth, and drought stress limits plant growth [4]. Notably, the *hd2ahd2b* mutant shows smaller rosette leaves in comparison to the WT, which may also contribute to the drought tolerance, because of the lower transpirational loss of water (Figure 7F). Additionally, small leaves achieve a higher leaf water-use efficiency or light-saturated photosynthetic rate under dry climates [99], consistent with that, *hd2ahd2b* plants with smaller leaves show higher F_v_/F_m_ value under drought stress than WT plants (Figure 7D,E). The *hd2ahd2b* mutant growth ratio deceased 7 days earlier compared with WT plants, whereas the leaf area of the *hd2ahd2b* mutant still increased up to 21 days after stopping watering (Figure 7F). These results indicate that the *hd2ahd2b* mutant could sense the water-deficiency conditions earlier than the WT plants, experiencing relatively milder dehydration stress and inducing appropriate growth adjustments to better cope with the drought conditions.

Based on our results, we propose a model for the regulatory function of HD2A and HD2B in coordinating drought stress response (Figure 10). 

Under drought conditions, the expression of HD2A and HD2B was enhanced [48] and functioned together to enhance the deacetylation of NCED9 and ABA-dependent/independent drought-responsive genes, thus repressing the endogenous ABA synthesis and drought responses. Additionally, drought stress increases the endogenous ABA content, which represses the expression of HD2A and HD2B, suggesting a negative feedback loop between ABA and HD2A/HD2B accumulation (Figure 9B). Furthermore, HD2A/HD2B may also be involved in drought regulation by affecting ribosome biogenesis. In sum, we showed that the Arabidopsis plant-specific histone deacetylases HD2A and HD2B have a redundant function and are involved in regulating the drought stress response through multiple pathways. Moreover, our data revealed a link between histone deacetylation and the activation of drought-responsive genes during the drought stress response.

It has been demonstrated that RPD3 or HDA1 histone deacetylases cannot discern and bind specific DNA sequences by themselves and are recruited to target genes promoters by DNA-binding proteins or associated co-repressors, stabilizing chromatin structure by removing histone acetyl from histone protein, thereby repressing transcription [100,101,102]. Similar to RPD3 and HDA1 histone deacetylases, the HD2 histone deacetylases HD2A and HD2B could be recruited by plant transcription factors to gene promoter region to mediate repression [57,103]. The important questions are how HD2A and HD2B are recruited and targeted to drought-responsive genes, which accessory transcription factors are involved, and how the transcription factor-HD2A/HD2B complex is integrated into a regulation mechanism during the drought stress response. HIGH-LEVEL EXPRESSION OF SUGAR-INDUCIBLE GENE2 (HSI2) and HSI2-like 1(HSL1) could recruit HD2A/HD2B to the DOG1 loci during seed germination, and HSI2 was considered to be a negative regulator of drought stress [104]. Moreover, analysis of the transcriptomes of *hd2ahd2b* mutant and *hsi2hsl1* mutant [105] revealed that about 33% (280 in 855) of the up-regulated genes in the *hsi2hsl1* mutant are also up-regulated in the *hd2ahd2b* mutant. Moreover, GO enrichment analysis demonstrated that genes of drought stress-related terms, such as “response to ABA”, “response to water deprivation”, and “response to osmotic stress” are enriched indicating that HSI2 and HSL1 might be involved in drought stress response as co-repressors of HD2A and HD2B. However, the synergistic interactions between HD2A/HD2B and HSI2/HSL1 during drought stress still need to be explored in detail.

The transcription of most genes is suppressed when cells move into mitosis [106], which is generally caused by the reduced histone acetylation levels largely prompted by the actions of HDACs [107,108,109]. Furthermore, deacetylation is important for accurate chromatin segregation [110]. Most histone acetylation could be reinstated rapidly by the histones when cell division is complete, which is important for cells to retain their characteristic gene expression patterns, and hence their identity, through multiple mitoses. That suggests cell-type-specific patterns of histone acetylation persist through the cycle. For the *hd2ahd2b* mutant, the leaves possess a clear serration compared with WT plants both in small seedlings and adult plants. That means the defective double mutant leaf is because of the abnormal formation of leaf primordia but not cell division or post-mitotic cell expansion in the leaf. Ueno et al. added that adaxial-abaxial leaf polarity to the disrupted levels and/or patterns of miR165/166 distribution in *hd2ahd2b* leaves [111]. In addition, the abnormal cotyledons in the early germinated *hd2ahd2b* seedlings indicate that cotyledon formation during embryogenesis is aberrated [52]. Based on these results, although the exact molecular roles of HD2A and HD2B in leaf morphogenesis is still largely unknown, we could reach the conclusion that HD2A and HD2B mainly function on the plant shoots apical meristem but not during post-mitotic cell expansion.

## 4. Methods

### 4.1. Plant Materials and Growth Conditions

The *Arabidopsis thaliana* ecotype Columbia-0 (Col-0) or mutants in the Col-0 background were used in all experiments. The T-DNA insertion lines GABI_355H03 (*hd2a*) and Sail_1247_A02 (*hd2b*), were described previously [53] and were verified by genome PCR using primers listed in (Appendix A). The double mutant *hd2ahd2b* was produced by crossing and homozygous lines were isolated by genotyping with primers listed in (Appendix A) and used for further analysis. The *pHD2B: HD2B-GFP* line is a complementation line in the *hd2ahd2b* background, and was a gift from Ton Bisseling [53]. Seeds were sown in moist soil mixed in a ratio 10:1 with sand and grown in the growth chamber under short-day conditions (10 h light/14 h dark and 20 °C/16 °C, respectively) for vegetative growth and long-day condition (14 h light/10 h dark and 20 °C/18 °C, respectively) for seed production, the light intensity in both conditions was ca. 100 to 130 µmol/sm^2^. The Arabidopsis plants used for seed production were grown first under short-day conditions for 4 weeks before being transferred to long-day conditions for flowering. 

### 4.2. Seed Germination, Root Growth, and Green Cotyledon Assay

The WT and mutant seeds were harvested at the same time and dry stored at room temperature in the dark for at least 1 month before being used for the subsequent test. Seeds used for the seed germination test were sown on water-saturated filter paper with 0 or 0.5 µM ABA and transferred into a growth chamber with long-day conditions without 4 °C treatment. The germination rates were assessed every 24 h during the incubation. For the root growth and green cotyledon assay, seeds of mutant and WT lines were sterilized with 75% (*v*/*v*) ethanol and sown on 1/2 MS plates (half-strength Murashige and Skoog (MS) solid medium, MS powder 2.35 g/L) with or without the indicated concentration of ABA, and then grown in a growth chamber with short-day conditions. Seedling root length and the ratio of green cotyledons were measured after 7 days of growth. For each independent experiment, approximately 100 after-ripened seeds were used for the seed germination test, and 30 seeds were used for root length and green cotyledon assay. The seed germination ratio was scored as “radicle emergence”. For each experiment, 3 replicates were evaluated.

### 4.3. Drought Treatment and Data Collection with High-Throughput Automated Non-Invasive Phenotyping Platform 

For the PEG-simulated drought, the well-developed 4 weeks old mutant and WT Arabidopsis plants growing under short-day conditions were irrigated with 25% PEG-6000 solution, and the pictures were taken 24h late after the treatment. For the nature drought stress, the seeds of A. th. Columbia-0 (Col-0) and *hda2ahd2b* double mutant line were sown into TAGLOK pots 68 mm^2^ (90 × 68 × 68 mm) containing 150 g well-watered soil and covered with transparent film to avoid rapid water loss. After being stratified for 3 days at 4 °C in the dark, the seeds were germinated and grown in a growth chamber and 14 days old seedlings were transferred to the PlantScreenTM Compact System in a controlled environment and drought stress was applied by withholding watering. In total, 60 plants for each line were employed and half of them were subjected to drought treatment and the other half to control. Control and drought-stressed plants were automatically phenotyped for RGB and kinetic chlorophyll fluorescence (ChlF) traits by the system. Before the data collection, the plants were transported to the dark acclimation chamber by conveyor belts for an initial 30 min dark adaptation. RGB images used for assessing plant development and morphological traits were taken with a UI-5580CP camera with a resolution of 2560 × 1920 pixels, and the total area covered by the plant (Area) was computed. To study the drought stress effect, the chlorophyll fluorescence was measured by FluorCam fluorometer, an imaging unit in the PlantScreen™ system. The plant PSII reaction centers were open after being treated for 30 min under dark conditions, the plants were transported to the chlorophyll imaging cabinet, and a flash of light was applied to measure the minimum level of fluorescence in the dark-adapted state (F_o_), followed by a saturation pulse used to determine the maximum fluorescence in the dark-adapted state (F_m_). The measurements were performed using an F_v_/F_m_ protocol. The pot weight was measured every 24 h automatically by the system.

### 4.4. Determination of ABA, NBT Detection of Superoxide, and Stomatal Aperture Measurement

The endogenous ABA content was measured with an Agilent 1290 Infinity II-6470 triple quadrupole LC/MS/MS System according to previously reported method [76] with a few modifications. Briefly, 0.5 g of fresh plant leaves were harvested and ground into a fine powder with liquid nitrogen and were transferred into a 2 mL microtube containing 1 mL of ethyl acetate. The samples were vortexed, followed by 30 min of shaking incubation at 4 °C, and then were centrifuged at 12,000× *g* for 10 min at 4 °C; the supernatant was transferred into a new 1.5 mL tube and evaporated to dryness using a vacuum concentrator (Eppendorf, Framingham, MA, USA). The residue was re-dissolved in 200 μL of 50% methanol and filtered through a 0.22 μm filter for sample loading. For each sample, 100 μL of methanol solution was subjected to LC-MS/MS analysis. ABA (yuan ye biotech, catalog no. B50724) was used based on authentic reference standards.

NBT staining was applied in accordance with a previously described method (Lee et al., 2002), and the 4-week-old wild type and *hd2ahd2b* plants grown under long-day conditions were subjected to PEG 6000 (25%) for 24 h to simulate drought stresses. Next, the unstressed and stressed wild type and the *hd2ahd2b* whole plants were immersed in 1 mg/mL fresh NBT solution (0.1 mg mL21 NBTin 25mM HEPES buffer, pH 7.6) for ROS staining. The plants were incubated in NBT staining solution and vacuumed at −0.1 MPa for 10 min, left for 1h at room temperature in the dark, and then the chlorophyll was bleached with 95% ethanol (80 °C), changing the 95% ethanol every 10 min. After the green color of the sample faded, a photograph of the leaves was taken and the content of superoxide was estimated using Image J, by quantifying the ratio of stained areas to whole areas. Next, the representative images were shown for both experiments. All determinations were made in triplicate.

Stomatal aperture measurements were performed as described previously with modification [83]. For stomatal aperture measurement, the 5th fully expanded 4 weeks old wild type and *hd2ahd2b* plants leaves were excised and immediately floated in the water for 2 h under light (100 to 130 μmol/s^−1^m^−2^) for inducing stomatal opening. Subsequently, the leaves were transferred to the ABA solution and PEG solution indicated for 10 min, the epidermis of the leaf abaxial was separated with taps and 20 stomatal apertures for each sample were measured. The inner edges of guard cells’ width, as well as the length of stomata, were measured by Image J. And stomatal density was also measured with the same epidermis.

### 4.5. DNA Extraction, RNA Extraction, cDNA Synthesis, and Gene Expression Analyses

The crude total RNA was extracted from developing Arabidopsis siliques, dry seeds, or imbibed seeds using Trizol reagent [112], and purified with RNeasy Plant Mini Kit (Qiagen, Cat No. 74904) according to the manufacturer’s instruction to remove genomic DNA contamination. 0.5 μg of total purified RNA was used for the cDNA synthesis with random hexamer primers using a QuantiTect Rev. Transcription Kit (Qiagen, Cat No. 205311). And the cDNA was diluted 50-fold with water for subsequent PCR. RT-qPCR was performed in a 20 μL reaction with 10 μL Sybr green (Bioline, Cat No. QT625-05), 1 μL of 10 μM specific primers, 5 μL dH2O, and 3 μL diluted cDNA. The thermocycling conditions were as follows: 10 min at 95 °C, followed by 45 cycles of 15 s at 95 °C, 15 s at 55 °C, and 45 s at 72 °C, and a dissociation step, 15 s at 95 °C, 60 s at 60 °C, and 15 s at 95 °C. RT–qPCR results were normalized using UBIQUITIN 5 (UBQ5) and S16 (At5g18380) as the internal control. The primers used for RT-qPCR are listed in (Appendix A). At least three biological replicates were analyzed. The relative expression levels of target genes were calculated with formula 2^−ΔΔCT^. Arabidopsis Genomic DNA has been extracted from 10 days seedlings with a modified CTAB method [113]. And 100 ng gDNA was used as a template for phenotyping the T-DNA insertion lines.

### 4.6. RNA-Seq Analysis

Arabidopsis WT and *hd2ahd2b* plants were grown under short-day conditions for 10 days. Ten-day-old WT and *hd2ahd2b* seedlings were treated with or without 100 μM ABA for 6h and sampled for RNA extraction. The total RNA of four biological replicates was extracted with an RNeasy Plant Mini Kit (Qiagen, catalog no. 74904) according to the manufacturer’s instructions. Single-end sequencing of all RNA samples was performed using two lanes of a NovaSeq SP system (Illumina, San Diego, CA, USA). Reads were aligned against the TAIR10 genome assembly [114] by HISAT2-2.1.0 [115]. The alignments were sorted using Samtools-1.8 [116] and expression quantification was done with Stringtie-1.3.4 [117] for the Araport11 genome annotation [114]. Differential expression was computed via the R package DESeq2, version 1.20.0 [118]. GO enrichment and multi-dimensional scaling analyses were performed in R [119].

### 4.7. ChIP-qPCR Assays

The ChIP assay was performed as previously described [120]. The chromatin was extracted from 10d *pHD2B: HD2B-GFP* seedlings. About 2 g seedlings were cross-linked in cross-linking buffer (400 mM sucrose, 10 mM Tris-HCl pH 8.0, 5 mM β-mercaptoethanol, 1% formaldehyde) by vacuuming for 10 min and stopped by adding an end concentration of 0.125 M glycine and vacuuming for another 5 min. the cross-linked plant materials were washed 2 times with ice water, dried with paper towels, and ground into powder in liquid nitrogen. The chromatin was extracted with 20 mL extraction buffer (400 mM sucrose, 10 mM Tris-HCl pH 8.0, 5 mM β-mercaptoethanol, Protease inhibitor), the homogenate was incubated on the rotation platform for 20min and pelleted by centrifuging at 4000 rpm for 25 min at 4 °C. the pellet was washed with nuclei wash buffer (20 mM Tris/HCl, pH 7.4, 25% glycerol, 2.5 mM MgCl2, 0.2% Triton x-100) and resuspended with 600 μL nuclei sonication buffer (50 mM Tris-HCl pH 8.0, 10 mM EDTA pH 8.0, 1% SDS, Protease inhibitor). The chromatin was sheared to 200–1000 bp by sonication, and after spinning for 10 min at 12,000 g and 4 °C, the supernatant was directly used for immunoprecipitation with specific antibodies. For anti-H3K9ac, anti-H4K5ac, and anti-H4ac analysis, the antibodies (anti- H4ac, anti-H4K5ac, anti-H3K9ac) were coupled to the magnetic protein G beads by incubating at 4 °C on a rotation platform overnight. Afterward, 100 μL sonicated chromatin was mixed with specific antibody-coupled magic beads and incubated overnight at 4 °C on a rotating platform. After the beads were sequentially washed 2 times with low salt buffer (0.1% SDS, 1% Triton X-100, 2 mM EDTA, 20 mM Tris-HCl pH 8.0, 150 mM NaCl), high salt buffer (0.1% SDS, 1% Triton X-100, 2 mM EDTA, 20 mM Tris-HCl pH 8.0, 500 mM NaCl), LiCl buffer (0.25 M LiCl, 1% NP-40, 1% Sodium Deoxychlorate, 1 mM EDTA, 10 mM Tris-HCl pH 8.0), and TE buffer (10 mM Tris-HCl pH 8.0, 1 mM EDTA). Finally, the chromatin was eluted with 400 μL elution buffer (1% SDS, 100 mM NaHCO_3_) incubating at thermo-block for 20 min at 500 rpm and 65 °C, and reverse cross-linked chromatin was performed at 65 °C for over 6 h after added 16 μL 5M NaCl. After being treated with proteinase K and RNase, DNA was purified by phenol-chloroform method, eluted with dH_2_O, and quantified for qPCR. For anti-GFP analysis, the GFP- Trap agarose beads were used instead of antibody-coupled magic beads.

### 4.8. Determination of Chl Content

The chlorophyll content was measured with the acetone extraction method according to the previously reported method [121]. About 0.1 g fresh WT and *hd2ahd2b* leaves were collected and ground quickly with the stick in a 1.5 mL tube, subsequently, 1 mL 80% acetone was added to the tube and stored in the dark until the samples were completely white, then clarified by centrifugation for 10 min at 12,000 g. Finally, the absorbance of the supernatant was measured at wavelengths 663 and 645 (A_663_ and A_645_) with a spectrophotometer. And the chlorophyll content was calculated using the Lichtenthalters and Arnons equations as follows:

Chlorophyll a (μg/mL) = −1.93 × A_646_ + 11.93 × A_663_ × volume/weight

Chlorophyll b (μg/mL) = 20.36 × A_646_ − 5.50 × A_663_ × volume/weight

Total chlorophyll (μg/mL) = 6.43 × A_663_ + 18.43 × A_646_ × volume/weight

Each experiment represented three biological replicates.

### 4.9. Bimolecular Fluorescence Complementation Assay

For the bimolecular fluorescence complementation assays, the ORF of HD2A, HD2B, NOP56, and FIB2 (without stop codon) were transferred into the pDONOR221 vector by Gateway Cloning and subsequently shifted into the pBiFCt-2in1-NN vector (LR reactions) according to the described [122]. Then, the constructs were transferred into Arabidopsis protoplasts by PEG transformation as described [123]. After incubation for 16 h to 24 h in the dark, the YFP fluorescence signal was monitored using a laser scanning confocal microscope (Leica TCS SP8 confocal).

## Figures and Tables

**Figure 1 genes-14-01199-f001:**
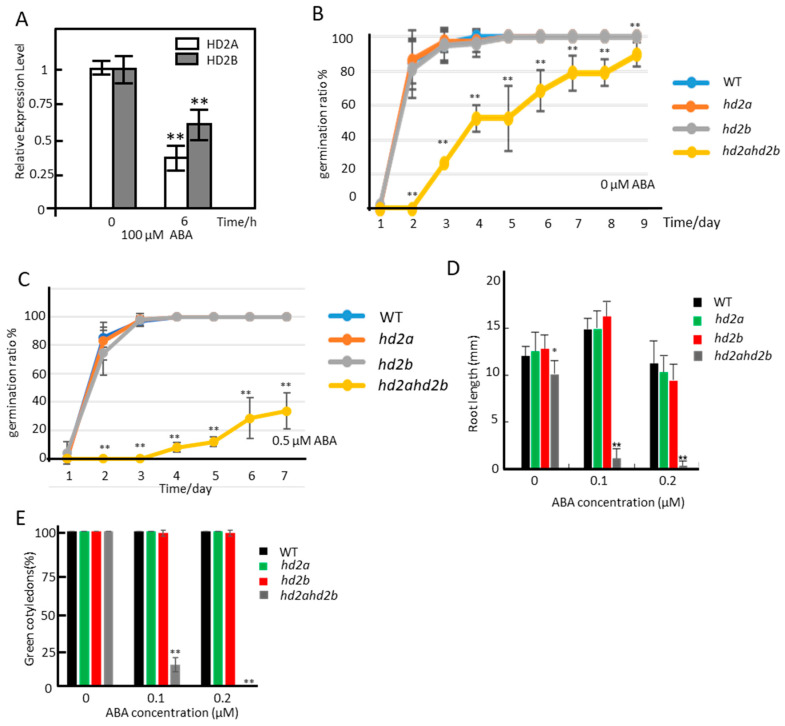
The *hd2ahd2b* plants show an ABA-hypersensitive phenotype. (**A**) Relative expression of *HD2A* and *HD2B* in 10-day-old seedlings after application of 100 µM ABA. UBQ5 served as an internal control. The seedlings were treated with 100 μM ABA by spraying. A total of 6 h later, the seedlings’ upper parts were collected for RNA analysis. Seedlings treated with water were used as control. (**B**,**C**) Seed germination efficiencies (radicle emergence) of WT, *hd2a*, *hd2b*, and *hd2ahd2b* in the presence of different concentrations of ABA. After-ripened seeds were imbibed on ½ MS plate in the presence of 0 and 0.5 µM ABA. The germination rate was scored according to the time course indicated. (**D**,**E**) Post-germination growth efficiencies of WT, *hd2a*, *hd2b*, and *hd2ahd2b* in the presence of different concentrations of ABA. Seeds of different lines were placed on ½ MS plates in the presence of different concentrations of ABA. The primary root length and cotyledon greening rate of 7-day-old seedlings were analyzed. All data represented are averages of three independent experiments (±SE). For each independent experiment, at least 90 seeds were used for seed germination analysis in (**B**,**C**), 20 seedlings were used for root length measurement in (**D**), and 90 seedlings were used for cotyledon greening rate analysis in (**E**). Asterisks indicate a significant difference between different lines (* *p* < 0.05, ** *p* < 0.01). One-way ANOVA (Tukey–Kramer test) analysis was performed.

**Figure 2 genes-14-01199-f002:**
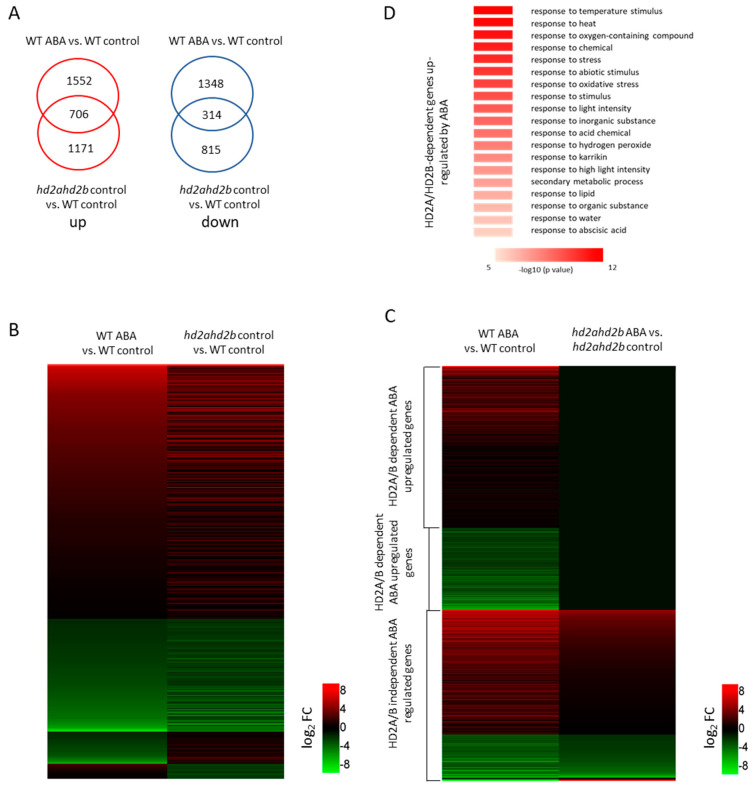
HD2A and HD2B positively regulate the expression of ABA-responsive genes. (**A**) Venn diagrams show the overlap of ABA-dependent, significantly up-regulated, and down-regulated WT genes with genes that are differentially expressed in the comparison of *hd2ahd2b* vs. WT control treatment. (**B**) Heat map showing the differences in expression levels of genes regulated by ABA in WT plants (left) and differentially regulated genes of *hd2ahd2b* vs. WT control treatment (right). (**C**) Heat map showing the differences in expression levels of genes regulated by ABA in WT (left) and *hd2ahd2b* (right) plants. (**D**) GO analysis of *HD2A*- and *HD2B*-dependent ABA up-regulated genes.

**Figure 3 genes-14-01199-f003:**
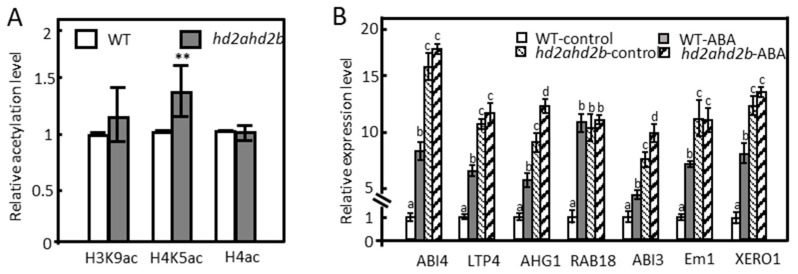
HD2B directly binds to and negatively regulates a subset of ABA-responsive genes. (**A**) The global histone acetylation levels of 4-week-old WT and *hd2ahd2b* plants. Histone modifications were analyzed by immunoblotting. The intensities of the signals of the immunoblot were quantified using the ImageJ software. Four biological replicates (mean +/− SD) were normalized to the signals of H4 (H4ac and H4K5ac) and H3 (H3K9ac). The WT signals were set to one for each analyzed histone modification. Asterisks indicate a significant difference between WT and *hd2ahd2b*. One-way ANOVA (Tukey–Kramer test) analysis was performed, (** *p* < 0.01). (**B**) RT-qPCR was performed to analyze the expression of selected genes in WT and *hd2ahd2b* plants in the presence or absence of 100 µM ABA (treatment for 6 h). *n* = 3, the values shown are means +/− SD. (**C**) ChIP-qPCR analysis of WT and *35Spro:: HD2B-GFP* complementation line. Immunoprecipitated DNA was obtained from 10-day-old WT and *35Spro:: HD2B-GFP* seedlings using an anti-GFP antibody. The relative amount of the indicated PCR products were quantified and normalized to internal control (s16). The values shown are means +/− SD of three biological replicates. Lowercase letters indicate significant differences (*p* < 0.05) between the different values. One-way ANOVA (Tukey–Kramer test) analysis was performed.

**Figure 4 genes-14-01199-f004:**
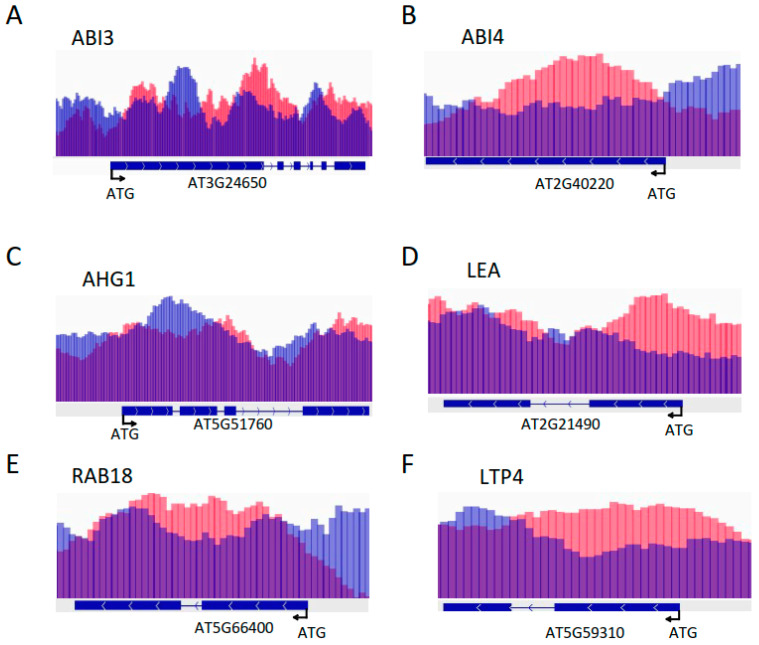
HD2A-GFP is enriched in ABA-responsive genes. Schematic illustration of ChIP-seq analysis of HD2A-GFP enrichment at the representative locus of different Arabidopsis gene loci: ABI3 (**A**), ABI4 (**B**), AHG1 (**C**), LEA (**D**), RAB18 (**E**), LTP4 (**F**), and XERO1 (**G**). Chip-seq data (Luo et al., 2021) were re-analyzed and displayed using the Integrative Genomics Viewer (IGV, Broad Institute). Red and blue tracks in the wiggle plot represent the normalized ChIP-seq read coverage (ChIP-seq input) and IP (carried out with anti-GFP antibody) chromatin sample (red tracks). The compared IGV tracks are overlaid and have the same scaling factor for the *y*-axis. Schematic representation of the corresponding genes and the translational start site (ATG) is highlighted.

**Figure 5 genes-14-01199-f005:**
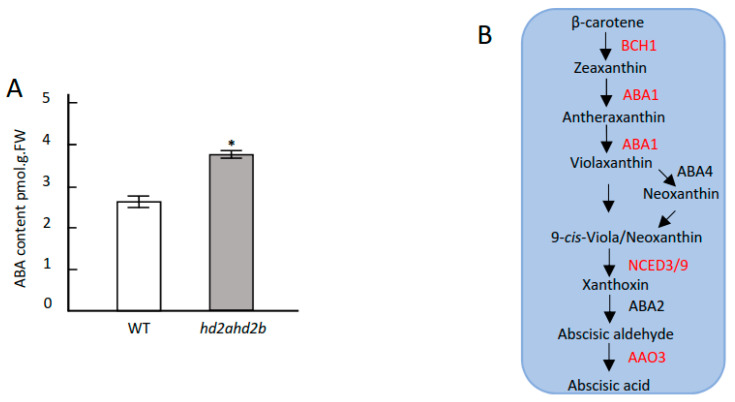
HD2A and HD2B negatively regulate ABA biosynthesis. (**A**) Endogenous ABA content in WT and *hd2ahd2b* plants. ABA content was determined in the leaves of four-week-old plants using LC-MS/MS. (**B**) Pathway of ABA biosynthesis. Analysed genes are highlighted in red. (**C**) The expression level of genes involved in ABA biosynthesis in WT and *hd2ahd2b* plants. RNA was prepared from four-week-old plant leaves and the expression level was determined using RT-qPCR. (**D**) H3K9 and H4K5 acetylation of AtABA1 and AtNCED9. The DNA of four-week-old WT and *hd2ahd2b* plant leaves was used for immunoprecipitation using anti-H3K9ac and anti-H4K5ac antibodies. (**E**) ChIP-qPCR analysis of *hd2b* plants complemented with 35Spro::HD2B-GFP. The DNA of ten-day-old WT and *pHD2B::HD2B-GFP* seedlings were used for immunoprecipitation with anti-GFP antibodies. The relative amount of PCR products were normalized to internal control (s16). (**F**) HD2A-GFP is enriched at the promoter and coding region of NCED9. For the schematic illustration of the NCED9 gene, the blue tracks indicate the input and the red tracks represent the IP (carried out with anti-GFP antibody) chromatin sample (red tracks). The compared IGV tracks are overlaid and have the same scaling factor for the *y*-axis. A schematic illustration of the corresponding genes and the translational start site (ATG) is shown below. The values shown are means +/− SD. Error bars represent the SD of three biological replicates for each experiment. Lowercase letters indicate significant differences (*p* < 0.05) between the different values. One-way ANOVA (Tukey–Kramer test) analysis was performed. (*: *p* < 0.05, **: *p* < 0.01).

**Figure 6 genes-14-01199-f006:**
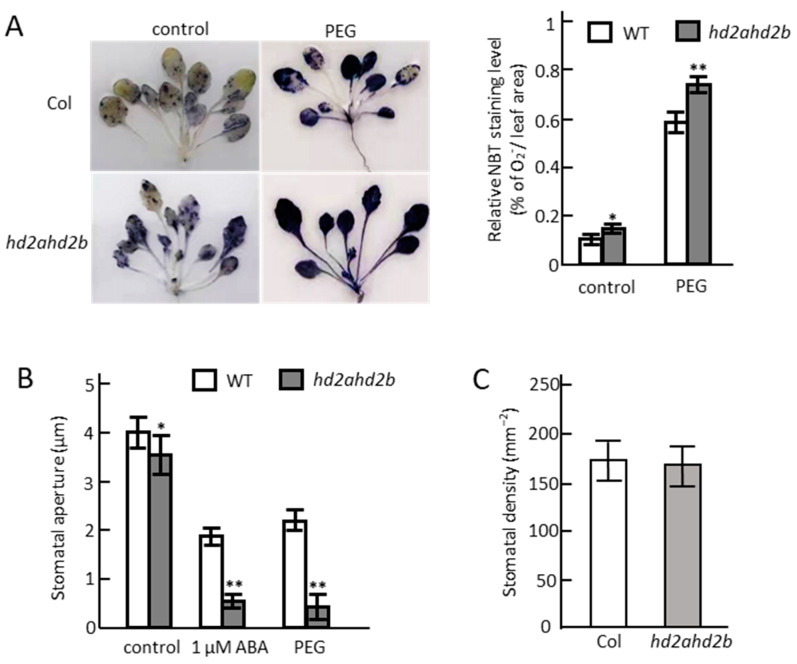
Loss of HD2A and HD2B function affects ROS accumulation and stomatal aperture. (**A**) ROS detection in WT and *hd2ahd2b* plants under control and drought stress conditions. Four-week-old WT and *hd2ahd2b* plants were subjected to PEG 6000 (25%) for 24 h to simulate drought stress conditions. Whole WT and *hd2ahd2b* plants were immersed in 1 mg/mL fresh NBT solution for ROS staining. After washing with ethanol, photographs were taken and the superoxide content was quantified with Image J. The relative NBT values represent the average (+/−SD) of thirty leaves for each line. (**B**) Effects of ABA and PEG on stomatal aperture in WT and *hd2ahd2b* leaves. Stomatal aperture was measured at the fifth fully expanded leaf of four-week-old WT and *hd2ahd2b* plants after treatment with 1 μM ABA and 25% PEG 6000 for 10 min. A total of 30 stomata per leaf of each line were analysed. (**C**) The stomatal density of WT and *hd2ahd2b* plants. Stomatal density was calculated at the fifth fully expanded leaf of four-week-old plants. One-way ANOVA (Tukey–Kramer test) (*: *p* < 0.05, **: *p* < 0.01). For (**B**,**C**), *n* = 3; for each replicate, the data were obtained from 30 stomata of each line. The vertical bar represents the mean and the error bars represent SEM.

**Figure 7 genes-14-01199-f007:**
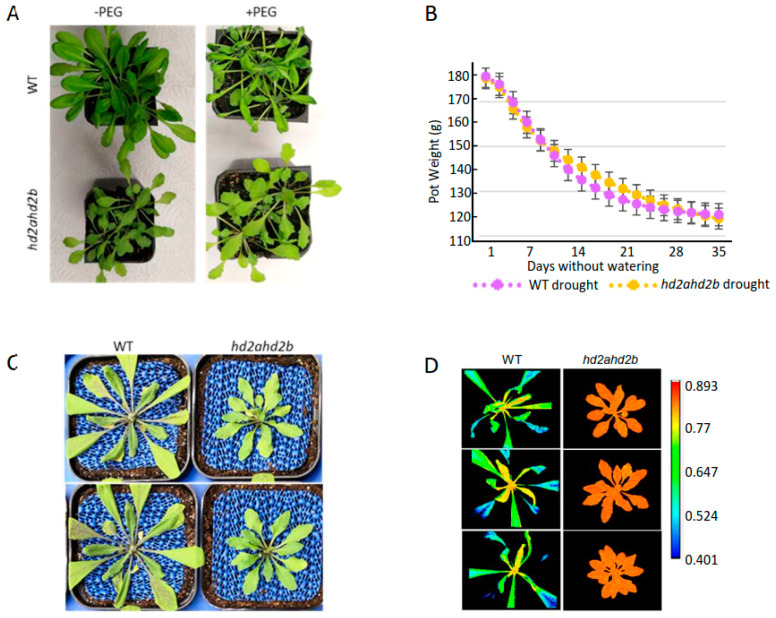
*hd2ahd2b* plants display enhanced drought tolerance in comparison to WT plants. (**A**) The drought-tolerant phenotype of *hd2ahd2b* plants. Four-week-old WT and *hd2ahd2b* plants were treated with 25% PEG-6000 to simulate drought stress conditions and photos were taken after 24 h. (**B**–**F**) Drought stress response was monitored using a PlantScreen^TM^ phenotyping system. (**B**) One-week-old WT and *hd2ahd2b* plants were transferred to the phenotyping platform. Watering was stopped and loss of soil water was monitored by weighing the pot every two days. (**C**) WT and *hd2ahd2b* plants after four weeks without watering. (**D**) Maximal PSII quantum efficiency (F_v_/F_m_) images of WT and *hd2ahd2b* plants after four weeks without watering. (**E**) Changes in maximal PSII quantum efficiency (F_v_/F_m_) and (**F**) leaf area of WT and *hd2ahd2b* plants during drought stress establishment. Values represent the average area of thirty plants for each line in (**B**,**E**,**F**). (**G**) Expression profiles of drought-stress-related marker genes in WT and *hd2ahd2b* plants under control and drought conditions. RNA was prepared from the leaves of four-week-old plants and the expression level was determined via RT-qPCR. UBQ5 served as an internal control. Lowercase letters indicate significant differences (*p* < 0.05) between the different values in (**G**). ANOVA and Holm–Sidak significance test were performed (*: *p* < 0.05, **: *p* < 0.01, ***: *p* < 0.001).

**Figure 8 genes-14-01199-f008:**
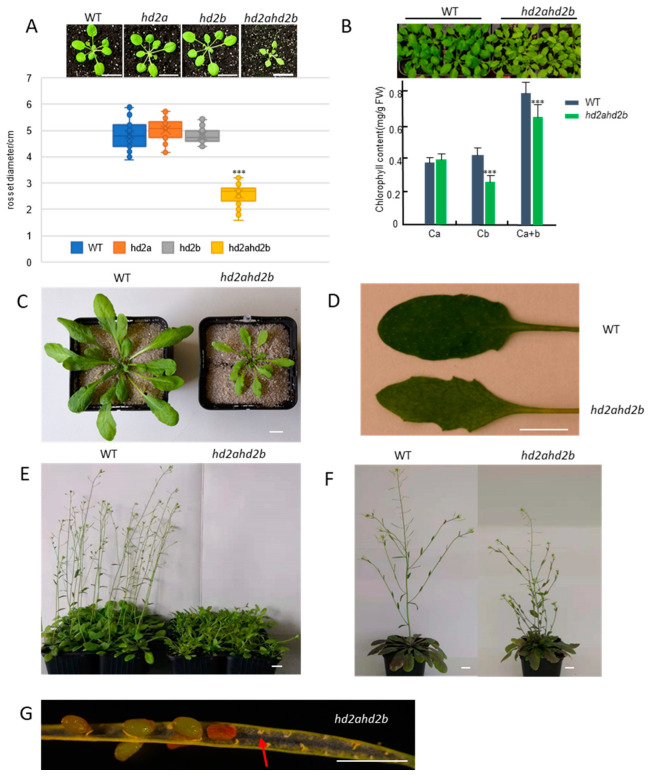
Phenotypes of *hd2a*, *hd2b*, *hd2ahd2b,* and WT plants. (**A**) Morphology of three-week-old WT, *hd2a*, *hd2b*, and *hd2ahd2b* plants. The *hd2ahd2b* double-mutant plants have smaller rosette diameters than the corresponding single mutants and the WT plants. (**B**) Chlorophyll b contents in *hd2ahd2b* plants are lower in comparison to WT plants. (**C**) Size of four-week-old *hd2ahd2b* and WT plants and (**D**) four-week-old rosette leaves of *hd2ahd2b* and WT plants. (**E**–**G**) The phenotype of reproductive elements of WT and *hd2ahd2b* plants. *hd2ahd2b* plants have a late-flowering phenotype (**E**) but generated more lateral shoots (**F**) and exhibited aborted seed development (**G**). The red arrow indicates unfertilized ovules. Asterisks in (**A**,**B**) indicate a significant difference from the WT in a Student *t*-test (*** *p* < 0.01). Scale bars = 2 cm (**A**), =1 cm (**C**), =1 cm (**D**), =1 cm (**E**), =1 cm (**F**).

**Figure 9 genes-14-01199-f009:**
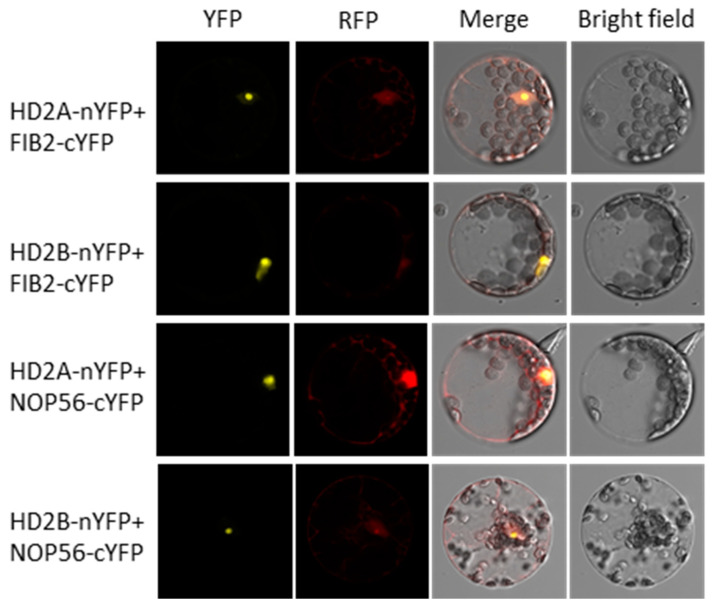
HD2A and HD2B interact with FIB2 and NOP56 in vivo. Bimolecular fluorescence complementation (BIFC) showing protein-protein interactions between HD2A, HD2B, FIB2, and NOP56 in Arabidopsis mesophyll protoplasts. HD2A and HD2B were fused to the N-terminus of Yellow fluorescent protein (nYFP), and FIB2 and NOP56 were fused to the C-terminus of Yellow fluorescent protein (cYFP). Both constructs were co-transfected into Arabidopsis mesophyll protoplasts as indicated and visualized using a confocal microscope after cultivating for 24 h at 25 °C. The red fluorescent protein (RFP) is an internal marker for transformation.

**Figure 10 genes-14-01199-f010:**
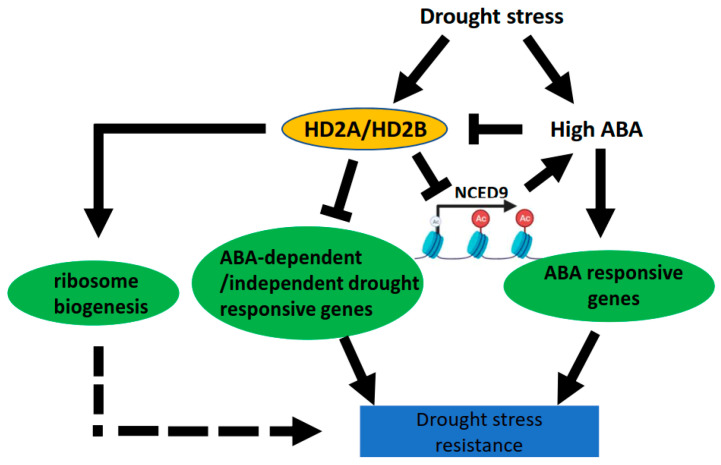
Schematic illustration of HD2A and HD2B function in drought stress. HD2A and HD2B repress ABA-dependent/independent drought-responsive genes. HD2A and HD2B are drought-associated genes active under drought stress conditions and repressed by ABA. HD2A and HD2B can downregulate ABA synthesis by repressing the ABA synthesis gene NCED9. Thus HD2A, HD2B, and endogenous ABA form a negative feedback loop.

## Data Availability

The data that support the findings of this study are available in the supplemental material of this article. The RNA sequencing data presented in the study are deposited in the ENA repository (https://www.ebi.ac.uk/ena/), accession number PRJEB62123.

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
