# Peer review of "Histone Deacetylases HD2A and HD2B Undergo Feedback Regulation by ABA and Modulate Drought Tolerance via Mediating ABA-Induced Transcriptional Repression"

_genes, 2023, doi:10.3390/genes14061199_

Round 1
Reviewer 1 Report
Reviewing comments on the article entitled « Histones deacetylases HD2A and HD2B undergo feed-back regulation by ABA and modulate drought tolerance via mediating ABA-induced transcriptional repression »
The authors present numerous experiments and analyses suggesting a role of two plant-specific deacetylases, HD2A and HD2B, as negative regulators during drought response. By the use of a Arabidopsis double mutant for these two deacetylases, they show a possible feed-back regulation by ABA and a modulation of drought tolerance through dependent and independent ABA pathways.
As a general comment, a lot of experiments are presented and constitute convincing evidences. Globally, the manuscript is correctly written and the experiments correctly explained with the exception of figure 5F and figure 9. The way to present bibliographic references should be revised and the use of certain terms should be homogenized.
As a conclusion on these comments, I recommend this article for publication but some important points should be corrected, rephrased and explained prior any full acceptance. Below is a list of these points:
Problems of figures:
Figure 1B and C: Why there’re no stars for significativity on these graphs?
Figure 2C: Only ABA-independent genes and ABA-dependent up-regulated genes are present in the legend on the side. What about ABA-dependent down-regulated genes? These genes should be presented too.
Figure 3: Why sometimes SD is used and sometimes is SE, for representing variation? The internal control s16 should be explained. What s16 stands for? There’s no explanation in the Mat&Met section where it’s written differently (S16).
Figure 4: The caption is not clear enough and the way to read these graphs should be better explained. The IGV acronym should be used just after the full terms and there’s a mistake about the TSS. The ATG codon is not a transcriptional start site but a translational one.
Figure 5: Same comment as for figure 4: Not enough explanation. In 5A graph, the unit presented on the side is wrong. Please check. What is FW? In 5D, how the quantification was performed? What is the input?
Figure 6A: The photo is too small, too dark and not well defined. It’s hard to see any difference. The quality of this picture should be improved (with lighter background for example).
Figure 8A: There’s an error in the legend on the side. It’s “Rosette” and not “Rosset”.
Figure 9: This figure should be fully revised. There’s no explanation about each column: DIC? Merge? Yellow fluo? Red fluo? The caption suffers from a lack of information too. No explanation about the different kind of fluorescence nor the kind of images presented.
Problems of scientific comprehension:
- About re-analysis of ChIP-seq data in figure 4, only a single sentence to describe this analysis is not enough (L305 to L307). This must be developed.
Furthermore, the conclusion should be moderated since not all ABA-responsive genes are repressed by HD2A and HD2B. I propose to rephrase L309: “…repressors of numerous ABA-responsive…”.
- About data presented in figure 5, there’s no reference at all to Fig. 5F in the text. Not a word. This should be added.
- About figure 7C, the description in the text is confusing. I propose to rephrase L449: “…green and dense, whereas those of the WT plants were etiolated and wilt.”
- L452: Fv/Fm instead of Fm/Fv.
- About interaction test with BiFC technic in figure 9, there’s no introduction of the interest to do this experiment. It seems to come from nowhere. The reason to test those interaction should be presented and explained.
Furthermore, the BiFC technic allows to localize the subcellular compartment where the interaction takes place. There’s no exploitation of this aspect of BiFC and no explanation about the red fluorescence. It seems to correspond to a nuclear marker but there’s no information in the text. This paragraph must be fully revised, as the corresponding figure.
- L525: Since response to abiotic stress is also a biological process, the word “biological” should be replaced by “developmental”.
- L547 to L551: 2 opposite results are presented: HD2A, B and D are positive regulator of drought tolerance (Tahir 2022) but the HD2D silencing confers a drought tolerance which means it acts as a negative regulator (Zhao 2022). There’s no discussion about this discrepancy and the authors claim that they found HD2A and B acting as negative regulator as expected. Why it was expected since there’re 2 different possibilities? This point must be clarified.
- L564 to L569: This paragraph correspond to an introduction and should be replaced at the beginning of this part (L534) not at the end.
- L592: The word “increases” alone in confusing because one would expect a decrease of expression in case of deacetylase activity. I propose to rephrase like that “… which decreases or increases the expression…”.
- L626 to L628: This sentence has no sense. It should be rephrased.
- L652: I don’t understand this sentence because claiming that “growing evidence suggests that translation process is crucial for organisms’ normal development” is just obvious: it’s a fact well established. This sentence should be rephrased.
- L657: What is “ER stress”?
- L678: The conclusion is poorly developed since there’s no mention of the involvement in stress response. I propose to add “…of Arabidopsis and involvement in stress response.”
L704: I don’t understand why HD2A and B counteract histone deacetylase. I was expected the opposite that is to say histone acetylase. This should be clearer.
- L719 to L723: This sentence has no sense. It should be rephrased.
Problems of uniformity:
There’s many different writing of the same terms all along the text. This should be homogenized.
For example: use 10-day-old instead of 10 days old.
When a word has been defined by its acronym for the first time, use this acronym throughout the text afterward.
For example: WT instead of wild-type or ABA instead of abscisic acid. Check all (there’s a lot).
The species names have to be written in italic with a capital on the first name.
L656: Saccharomyces cerevisiae instead of saccharomyces cerevisiae.
Latin words have to be written in italic.
For example: in vivo in L513.
Problems of referencing:
Some bibliographic citations are not correct.
L95 (2 parenthesis?), L139, L599, L615, L856.
Check all.
Problems of references:
Some articles in the references section are not well cited.
L945: The journal name is missing
L1004: The journal name is missing
L1006: The pages are missing
L1008: The journal name and pages are missing
L1019: The journal name is missing
L1024: The way to cite this reference is not correct
I didn’t check all the references but it should be done more carefully for all.
Problems of tense, grammar or syntax:
Sometimes the past is used instead of present and vice-versa.
L53: catalyzes instead of catalyzed
L63: is instead of was
L93: the word “and” has to be eliminated
L225: the word “were” has to be eliminated
L304: wanted instead of want
L457: analyzed instead of analyze
L550: silencing instead of silence
L556: compared to WT instead of than wild-type
L577: hypersensitivity instead of hypersensitive
L650: translation instead of translational
L656: the word “in” is missing between deletion and Saccharomyces
L682: the second word “loss” has to be eliminated
L686: decreased instead of decease / earlier instead of early
L708: figure instead of figur
Miscellaneous:
L91: the use of the word “Recently” for a citation of a work done 15 years ago is not correct.
As I mentioned previously, the global english quality is correct but needs some improvement. I detailled things to edit in the "Problems of tense, grammar or syntax" section in my review.
Author Response
"Please see the attachment."

Reviewer 2 Report
The manuscript investigates the molecular relationship between plant-specific histone deacetylase subfamily HD2s and abscisic acid (ABA) during the vegetative phase in Arabidopsis. The manuscript presents a well-designed study with clear and concise descriptions of the methods and results. The authors provide ample evidence to support their conclusions, and the data presented is well-organized and analyzed. The manuscript is well-written, and the flow is excellent. However, the introduction could be expanded to provide more background information on drought with recent references. The authors could also provide more information on the biological significance of the observed changes in gene expression and histone acetylation levels. However, the introduction could be expanded to provide more background information on drought with some recent references. The authors could also provide more information on the biological significance of the observed changes in gene expression and histone acetylation levels. The below comments should be addressed before going further.
Some comments need to be addressed before going further. For instance, the author could describe more about developing deeper and larger root systems, reducing water loss by regulating stomatal closure, and increasing the level of antioxidants for drought stress with recent references. Below are detailed comments:
L36-42. The author should describe more about the development of deeper and larger root systems, reducing water loss by regulating stomatal closure, and increasing the level of antioxidants for the drought stress with the more recent ref as well as in ABA signaling pathways. such as https://doi.org/10.3390/ijms21030996.
L59-62, The recent papers about ABA component PP2C with alternative splicing variants should be included here (https://doi.org/10.3389/fpls.2022.851531)
L91-93, Tanaka et al.2008 might not be a recent study
L117, when ROS and PEG firstly appear, please indicate the full name
L139, the ref style here might not be consistent with other refs.
L168, please revise the words “interaction”; it is confused with protein-protein interaction
L174, keep all consistent with (A) not A)
L266, it is so confused to come up with ChIP-qPCR results. Did the author do ChIP? These
important experiments should be mentioned in the main text. Later find it in L297.
L309, no experiments can support this conclusion: via the removal of histone acetylation at these gene loci
L314-322, the color code is not clear here.
L330, Figure5 the front is not consistent
L512, where is these interaction proteins come from? It seems the loss of connection with the whole manuscript, seems better not to have it here.
L558-559, Usually, increasing root water uptake from the soil and reducing water loss by closing stomata are the two main ways for plants to cope with drought stress, the recent ref. should be included here: https://doi.org/10.3390/ijms23116025
English is fine and minor revision is needed
Author Response
"Please see the attachment."
